# PRe-hospital Evaluation of Sensitive TrOponin (PRESTO) Study: multicentre prospective diagnostic accuracy study protocol

Abdulrhman Alghamdi ![ORCID],[1,2] Eloïse Cook ![ORCID],[3] Edward Carlton,[4] Aloysius Siriwardena ![ORCID],[5] Mark Hann,[6] Alexander Thompson,[6] Angela Foulkes,[7] John Phillips,[8] Jamie Cooper,[9] Steve Bell,[10] Kim Kirby ![ORCID],[11] Andy Rosser,[12] Richard Body[1,3]

For numbered affiliations see end of article.

**Correspondence to**
Mr Abdulrhman Alghamdi;
abdulrhman.alghamdi@mft.nhs.uk

## ABSTRACT

**Introduction** Within the UK, chest pain is one of the most common reasons for emergency (999) ambulance calls and the most common reason for emergency hospital admission. Diagnosing acute coronary syndromes (ACS) in a patient with chest pain in the prehospital setting by a paramedic is challenging. The Troponin-only Manchester Acute Coronary Syndromes (T-MACS) decision rule is a validated tool used in the emergency department (ED) to stratify patients with suspected ACS following a single blood test.

We are seeking to evaluate the diagnostic accuracy of the T-MACS decision aid algorithm to 'rule out' ACS when used in the prehospital environment with point-of-care troponin assays. If successful, this could allow paramedics to immediately rule out ACS for patients in the 'very low risk' group and avoid the need for transport to the ED, while also risk stratifying other patients using a single blood sample taken in the prehospital setting.

**Methods and analysis** We will recruit patients who call emergency (999) ambulance services where the responding paramedic suspects cardiac chest pain. The data required to apply T-MACS will be prospectively recorded by paramedics who are responding to each patient. Paramedics will be required to draw a venous blood sample at the time of arrival to the patient. Blood samples will later be tested in batches for cardiac troponin, using commercially available troponin assays. The primary outcome will be a diagnosis of acute myocardial infarction, established at the time of initial hospital admission. The secondary outcomes will include any major adverse cardiac events within 30 days of enrolment.

**Ethics and dissemination** The study obtained approval from the National Research Ethics Service (reference: 18/ES/0101) and the Health Research Authority. We will publish our findings in a high impact general medical journal.

**Trial registration number** Registration number: ClinicalTrials.gov, study ID: NCT03561051

## Strengths and limitations of this study

► Prehospital Evaluation of Sensitive Troponin is a multicentre prospective observational diagnostic accuracy study recruiting patients from four ambulance services in the UK, so we anticipate that our result truly reflects UK practice.

► The future clinical use of Troponin-only Manchester Acute Coronary Syndromes in the prehospital setting will be limited due to the observational study design pending a definitive randomised controlled trial.

► The study captures a large amount of data which allow the study team to evaluate different emergency department strategies used to risk stratify patients with chest pain in the prehospital setting.

► The study evaluates three different point-of-care troponin assays in conjunction with multiple validated decision aids.

## INTRODUCTION

Chest pain is one of the most common reason for emergency hospital admission. Clinicians will suspect a diagnosis of acute coronary syndromes (ACS) in approximately half of the patients presenting to emergency departments (EDs) with chest pain, accounting for the majority of these admissions. However, less than 20% of those admitted to hospital on the suspicion of ACS actually have that diagnosis. Most of these admissions could be avoided with improved diagnostic technology.[1–3]

In recent years, there has been much work performed in the ED setting with the aim of rapidly risk-stratifying patients with cardiac chest pain with a view to early discharge of those who are at low risk. The Troponin-only Manchester Acute Coronary Syndromes (T-MACS) is a scientifically derived mathematical model that combines clinical and historical features with ECG and cardiac biomarker results to determine the probability of ACS and assign patients to one of four risk groups: very low risk (<2% probability), low risk

(2%–5% probability), moderate risk (5%–95% probability) and high risk (probability ≥95%).

T-MACS has been shown to effectively reduce unnecessary hospital admissions when used in the ED.[4] It identifies 45% of patients as eligible for safe, immediate discharge following a single blood test.[4 5] This is demonstrably superior to other early rule-out strategies, including that recommended by National Institute for Health and Care Excellence (NICE) prior to publication of our findings. In addition to 'ruling out' a diagnosis of ACS and reducing the need for unnecessary investigations and hospital admission, T-MACS can also 'rule in' the diagnosis in approximately 5% of patients with 95% positive predictive value, facilitating early specialist treatment.[4–7]

The original T-MACS model relies on laboratory-based troponin testing. In the multicentre Bedside Evaluation of Sensitive Troponin (BEST) study, we have evaluated the accuracy of T-MACS with point-of-care (POC) troponin assays that use portable/handheld analysers, which could be used in ambulances. Results from the BEST study demonstrate that, with a contemporary POC test, T-MACS 'ruled out' ACS in 42.7% of patients with 95.5% sensitivity and 98.7% negative predictive value.[5]

Using portable POC tests, T-MACS could remove the requirement for many patients with suspected ACS to be assessed in the hospital, enabling even earlier reassurance for patients and cost savings for the National Health Service (NHS). Given the high prevalence of chest pain, avoiding ED attendances will reduce crowding, which leads to more patient safety incidents and higher mortality. Similarly, avoiding unnecessary transfer to hospital will free up ambulances to answer other emergency calls. However, as blood tests will be taken sooner after symptom onset, it is not safe to assume that T-MACS will be accurate in the prehospital environment. We must formally evaluate its accuracy in that setting.

## Aims and objectives

The primary objective of Pre-hospital Evaluation of Sensitive Troponin is to evaluate whether paramedics can use troponin testing by a handheld device with a computerised algorithm at the time of arrival to patients with symptoms that cause the treating paramedic to suspect the diagnosis of ACS. This would avoid unnecessary transfer and enable accurate identification of: (1) patients who do not have acute myocardial infarction (AMI) and therefore do not need to be taken to hospital; and (2) patients who do have AMI and therefore need to be given early treatment in the prehospital environment for their condition. Secondary objectives include validating the T-MACS decision aid that could be used to enhance the early diagnosis of AMI.

## METHODS AND ANALYSIS

### Design and setting

We will undertake a multicentre, prospective diagnostic accuracy study involving ambulance services and EDs in the UK. We started participants recruitment on February 2019 and we will complete recruitment within 12 months.

Within each ambulance service, research activity will be focused at ambulance stations or hubs with (1) a track record for successful delivery of similar research; and (2) which feed into hospitals with (a) adequate central laboratory support for sample processing and storage, and (b) clinical protocols that adhere to national and international standards for the investigation of patients with suspected ACS.

### Data collection

#### Prehospital environment

Paramedics will record basic clinical data using a brief case report form at the time of inclusion. The data collected will be sufficient to enable calculation of the T-MACS decision aid outcome, including the treating paramedic's interpretation of the patient's 12-lead ECG. However, in this observational study, the T-MACS rule outcome will not be known to paramedics.

Paramedics will receive bespoke study training before signing the signature log. After participating paramedics have provided any necessary urgent treatment and obtained verbal consent, they will undertake venepuncture prior to transferring the patient. If paramedics are inserting an intravenous cannula, blood will be drawn at the same time. Less than 5 mL venous blood will be drawn and stored in a lithium heparin bottle labelled with a unique study identifier. The date and time of venepuncture will be logged on the case report form and on the blood bottle. All patients will then be transferred to hospital in accordance with routine care.

#### Hospital environment

On arrival at the hospital, all patients will undergo reference standard troponin testing in accordance with contemporary national and international guidance. Acceptable protocols for reference standard troponin testing include:

- If a contemporary (not high sensitivity) troponin assay is used: laboratory-based troponin testing on arrival and either 6 hours after arrival, or 10–12 hours after the onset of peak symptoms.
- If a high sensitivity troponin assay is used: laboratory-based troponin testing on arrival and either 3 hours after arrival, or 6 hours after the onset of peak symptoms, unless the patient has undergone investigation according to a validated rule-out protocol as advocated in national international guidelines.

A high-sensitivity troponin assay is defined as an assay that can detect troponin concentrations in at least 50% of apparently healthy individuals with a co-efficient of variation of <10% at the 99th percentile cut-off.

When the patient arrives at the ED, the local study team will be informed. A member of the study team will either send the lithium heparin sample drawn in the prehospital environment up to the central laboratory for processing and storage, or the sample will be processed and stored

by a member of the study team who has had appropriate training. Once the patient has received all initial treatment in accordance with routine care, a member of the local research team will then approach them to answer any questions they may have about the study and obtain written informed consent.

If the research team cannot obtain written informed consent at the time of participant admission to the ED, the research paramedic will follow this up and obtain written informed consent via post, electronically or over the phone. The research paramedic will have 4 weeks to obtain written informed consent before the participant is withdrawn from the study and their samples destroyed. The participant will be made aware of this deadline in the information that is sent to them before they are approached for consent. If the participant is followed up via postal consent, a 1-week grace period will be given to allow for delays in the postal service. After this, the participant will be withdrawn from the study and their study data and samples will be destroyed.

One hour (±30 min) after the prehospital blood has been taken; a member of the research team will draw another sample of venous blood (<5 mL) into the provided lithium heparin bottle labelled with the participant's unique study identifier. The date and time the blood was drawn will be logged on the case report form and on the blood bottle. This venous blood sample will be sent up to the central labs for processing and storage or will be processed and stored by the local research team.

### Follow-up
Patients will be followed up by reviewing clinical records relating to their inpatient course, including data from serial troponin testing; other laboratory analyses; length of stay; all imaging investigations and procedures and details of any haemorrhagic complications. We will also contact the participant's primary care practitioner after 60 days to obtain information about any additional relevant events occurring within 30 days of the initial ambulance call. In the small percentage of cases where participants do not have a primary care practitioner, we will contact participants directly after 30 days. If this is not possible and the participant is lost to follow up, then this will be recorded on the electronic case report form.

### Resource use
We will collect comprehensive data about secondary healthcare resource use at baseline and 30 days, which may be used to subsequently develop a cost-effectiveness model. Total direct healthcare costs will be identified and quantified according to the UK NHS perspective relevant to decision-makers within the NHS.[8] At baseline, data will be collected with regards to the initial ambulance call, such as the date and time of the call, the time of ambulance dispatch to the patient, the time of arrival to the patient and whether a rapid response unit was dispatched. Resource-use data collected at 30 days will include: time (hours) and length (days) of hospital stay

(total; on coronary care, high dependence and intensive care units); laboratory, radiological and cardiological investigations during the initial hospital stay; nature and duration of any procedures or cardiac surgery; management of haemorrhagic complications; details of admissions and further ED attendances. Data on resource use will be collected using structured data collection forms from patient medical records and supplemented by information obtained from the patient's primary care practitioner at follow-up.

### Sample processing
On arrival at the destination hospital, the labelled whole blood sample that was drawn in the prehospital environment for research will either be sent to the hospital laboratory for processing and storage along with study-specific instructions, or this will be carried out by members of the local research team. The local laboratory personnel/research team members will test the whole blood for POC troponin using the Roche cobas h 232 TnT and the leftover will be centrifuged to separate out the plasma. The plasma will then be stored in separate aliquots and transferred to the freezer within 8 hours of collection, pending subsequent analysis in batches. The relevant manufacturers of commercially available assays have verified sample stability under these conditions.

The laboratory/research team will process a second lithium heparin sample, drawn by the research nurse, 1 hour (±30 min) after the initial prehospital blood draw. For this sample, the local laboratory personnel/research team will centrifuge the blood sample and the plasma will then be divided into separate aliquots. This will then be stored in the freezer within 8 hours of collection. As above, samples will be stored pending subsequent analysis in batches.

Plasma will later be analysed for POC troponin assays, as follows: Abbott i-Stat troponin I and LumiraDx troponin I by the central study team. Leftover plasma will continue to be stored at the central study site to permit evaluation of additional, new POC troponin assays when developed.

### Participant selection
We will prospectively approach patients who have called for an emergency (999) ambulance with symptoms that the attending paramedic suspects may have been caused by an ACS.

#### Inclusion criteria
► Adult patients (>18 years).
► Called 999 for an emergency ambulance because they have experienced pain, discomfort or pressure in the:
  – Chest.
  – Epigastrium.
  – Neck.
  – Jaw.
  – Upper limb without an apparent non-cardiac source (compatible with the American Heart Association case definitions).[9]

► Attending paramedic suspects these symptoms may be caused by ACS.

### Exclusion criteria
► Patients with unequivocal evidence of ST-elevation myocardial infarction who are being immediately transferred for primary percutaneous coronary intervention.
► Patients in whom an alternative diagnosis (other than ACS) is suspected, which would necessitate transfer to hospital.
► Patients who have not experienced symptoms in the previous 24 hours.

Patients who are unable to provide written informed consent, either because they lack the mental capacity to provide written informed consent or because effective communication is not possible (eg, non-English speakers in the absence of adequate translation services).

### Sample size
The specificity of T-MACS is approximately 45%[10] and the prevalence of the primary outcome in this cohort will be approximately 10%. Assuming that we identify an algorithm with 100% sensitivity and negative predictive value, the lower bound of the 95% CI would be>90% for sensitivity and>99% for negative predictive value with a sample size of 605 participants. Accounting for potential loss to follow-up and missing data (~5%–10% based on experience in previous similar studies), we plan to include a total of 700 participants.

### Participant withdrawal
If the participant gives verbal consent for the paramedic to proceed with the blood sample, but loses capacity before the blood is drawn, the participant will be withdrawn from the study. All data and samples collected up to this point will be destroyed as we will be unable to obtain written informed consent. In the event that written informed consent cannot be obtained from the participant for any other reason, the participant will be withdrawn from the study. As above, all data and samples collected up to this point will be destroyed.

In the event that a patient who has given written informed consent loses capacity before the 30-day follow-up, the participant will be withdrawn from the study. Any identifiable data or tissue collected up to this point would be retained and used in the study as written consent had been given. The participant would not be followed up at 30 days.

If at any time, the study team believes that remaining on the study is not in the participant's best interest, they will approach the participant directly to discuss withdrawal from the study. However, if their withdrawal is recommended from their primary caregiver or a relative due to psychological distress or a similar reason, the study team will not seek to contact the participant any further and they will be withdrawn from the study. Any identifiable data or tissue collected up to this point would be retained and used in the study as consent had been given.

### Outcomes
The primary outcome will be a diagnosis of AMI, established at the time of initial hospital admission. To diagnose AMI according to internationally accepted standards requires serial troponin sampling. This will help to ensure adequate reference standards for the diagnosis of AMI. Outcomes will be adjudicated by two independent investigators with reference to relevant clinical information but blinded to the results of research investigations. Discrepancies will be resolved by a third independent investigator. AMI will be defined according to the Fourth Universal Definition.[11] By virtue of the inclusion criteria, all patients will have symptoms and signs consistent with myocardial ischaemia. Briefly, therefore, patients will be deemed to have met this outcome if they develop a rise and/or fall of troponin to above the 99th percentile.

The secondary outcomes will include any major adverse cardiac events, which include cardiovascular death, coronary revascularisation and incident AMI within 30 days. All causes of death occurring within 30 days and the final diagnoses of all patients will also be recorded.

### Statistical analysis
#### Primary analysis
We will determine the output of the T-MACS decision aid using the original, predetermined algorithm. This algorithm computes the probability that each patient has a diagnosis of ACS and stratifies patients into four groups on the basis of that probability, as follows: very low risk (<2% probability)—ACS can be considered ruled out. We hope our findings will justify avoiding transport to hospital in this group; low risk (2%–5% probability): this group requires serial troponin sampling, which could be taken in an ambulatory care setting; moderate risk (5%–95% probability): this group requires serial troponin sampling but may also require additional imaging and therefore requires transfer to hospital; high risk (≥95% probability): ACS is 'ruled in' for this group, which could facilitate direct transfer to a specialist centre, facilitating early coronary intervention.

We will calculate the diagnostic accuracy of T-MACS as a 'rule-out' tool by dichotomising the probability at a threshold of 2%. We will then calculate sensitivity, specificity, positive and negative predictive values, positive and negative likelihood ratios and their respective 95% CIs. We will also report the number and percentage of patients with ACS stratified by T-MACS risk group. The proportion of transfers to the ED that could have been avoided will be calculated.

#### Secondary analyses
We are planning to conduct several secondary analyses. First, we will evaluate the diagnostic accuracy of T-MACS as a 'rule-in' tool, which would facilitate direct transfer to tertiary care heart attack centres and may enable patients to benefit from earlier specialist treatment such as

percutaneous coronary intervention in future. To do this, we will dichotomise T-MACS at the probability threshold of 95% (ie, analysing classification as 'high risk' vs all other risk groups). We will again calculate sensitivity, specificity, positive and negative predictive values, positive and negative likelihood ratios and their respective 95% CIs. Finally, we will calculate the proportion of patients with and without ACS who would have been transported to tertiary care facilities if T-MACS had been used in practice, and we will compare those findings to the observed practice in routine care. As some patients may have final diagnoses other than ACS, we will also retrieve the final coded diagnosis for all patients and present a descriptive analysis stratified by T-MACS risk group.

Second, we will evaluate the diagnostic accuracy of other hospital-based strategies used to rule-out, rule-in or 'risk-stratify' patients with AMI or ACS such as the HEART (History, ECG, Age, Risk factors, Troponin) score, History and ECG only Manchester Acute Coronary Syndromes (HE-MACS) decision aid, limit of detection strategy and selected cut-offs (eg, the 99th percentile of a reference population).

### Economic analyses
Total direct healthcare costs (resource use × unit costs data) will be calculated using a microcosting study run by our team (ISRCTN 86818215) and will compare the novel diagnostic pathway versus estimates for current care.[12] Where appropriate, we will proceed to formal cost-effectiveness analysis using a de novo decision-analytic model populated with data collected during this study and other externally published data. The model would extrapolate the effects of implementing the novel diagnostic pathway derived in this work on healthcare resource use and health status (quality adjusted life years as informed by the EQ-5D) versus current diagnostic and treatment pathways. Economic analyses will be led by AT.

### Patient and public involvement
To maximise the potential for clinical impact, this study has been designed in collaboration/consultation with a rounded group of stakeholders including patient and public representatives (eg, the Withington Heart Help Group and the Ticker Club) and industry (we have consulted with numerous manufacturers to confirm that work in this area is currently a high priority). Manchester University NHS Foundation Trust sponsors the study. Our consent procedure has been informed by prior experience within our national network of ambulance services, initial feedback from patient and public representatives (who have agreed with concern that judgement may be clouded in this acute situation) and the experience of AS in the Wellcome-funded 'Network Exploring Ethics in Ambulance Trials (NEAT)' project.[13]

### DISSEMINATION
Following completion of our analysis, we will discuss the significance of our findings and the key messages

to be communicated at meetings of the Trial Steering Committee, Trial Management Group and Patient Advisory Group. Following this, we will finalise our dissemination strategy. We will aim to publish our findings (positive or negative) in a high impact general medical journal with a relevant target audience (eg, British Medical Journal; The Lancet). In addition, we aim to present our findings to relevant target audiences at national and international conferences (eg, Royal College of Emergency Medicine Annual Scientific Conference; European Society of Cardiology Annual Conference).

If our findings are positive, we will also develop an implementation strategy. This will involve working with commissioning groups (including NHS England and the Greater Manchester Joint Commissioning Board), NICE and Ambulance NHS Trusts to make the case for the clinical and cost effectiveness of our technology. Template clinical guidelines and training guides will be disseminated to ambulance services, and we anticipate proceeding to support pilot evaluations with a view to larger scale clinical implementation within 2 years.

Finally, we will work with stakeholder organisations including the National Ambulance Research Steering Group (NARSG; AS is a member) and the International Federation of Clinical Chemistry Committee for Cardiac Biomarkers (RB is a member) to enhance communication of our findings within the field.

### DISCUSSION
Based on our experience with previous studies, if our findings are positive we will aim to achieve clinical implementation within 2 years. Clearly, this will involve additional work to demonstrate the feasibility and acceptability of 'live application' of T-MACS in the ambulance; to develop new clinical guidelines and training regimes and to robustly communicate the clinical and cost effectiveness of the strategy.

The recent update to NICE Guideline CG95 incorporated a novel diagnostic strategy (originally developed by our group) for in-hospital use based on data from observational studies with a similar design. Given that precedent, we anticipate that our findings will generate the evidence required by NICE to issue a recommendation for the clinical use of T-MACS with a POC troponin assay in the prehospital environment.

We also implemented T-MACS in the hospital environment primarily based on observational data. The algorithm has been applied in 8000 patients and has led to 2/3 patients being safely treated in an ambulatory care environment without requiring hospital admission. Health Innovation Manchester, with access to a Joint Commissioning Board, has adopted T-MACS as an exemplar project for rapid implementation across Greater Manchester. We will conduct a 'phase 4 evaluation' of that regional implementation, aiming to achieve more widespread clinical implementation within 24 months of completion. If the findings of the proposed study are

positive, we will use a similar methodology to achieve rapid implementation in the prehospital environment.

If successfully implemented in practice, we anticipate that our findings will avoid the need for unnecessary ambulance transfers and hospital admission in approximately 40% of patients. As chest pain is the second most common reason for emergency ambulance calls and most common reason for emergency hospital admission, this is likely to have a substantial economic impact while reducing hospital overcrowding and its associated complications.

**Author affiliations**
[1]Division of Cardiovascular Sciences, University of Manchester, Manchester, UK
[2]College of Applied Medical Sciences, King Saud bin Abdulaziz University for Health Sciences, Riyadh, Saudi Arabia
[3]EMERGING Research Team, Manchester University NHS Foundation Trust, Manchester, UK
[4]Emergency Department, North Bristol NHS Trust, Westbury on Trym, UK
[5]School of Health and Social Care, University of Lincoln, Lincoln, UK
[6]Division of Population Health, The University of Manchester, Manchester, UK
[7]HeartHelp Support Group, Manchester, UK
[8]The Ticker Club, Manchester, UK
[9]Emergency Department, Aberdeen Royal Infirmary, Aberdeen, UK
[10]North West Ambulance Service NHS Trust, Bolton, UK
[11]South Western Ambulance Service NHS Foundation Trust, Plymouth, UK
[12]West Midlands Ambulance Service NHS Foundation Trust, Brierley Hill, UK

**Acknowledgements** This paper presents independent research funded by the National Institute for Health Research (NIHR) under its Research for Patient Benefit (RfPB) Programme (Grant Reference Number PB-PG-1216-20034). The views expressed are those of the author(s) and not necessarily those of the NIHR or the Department of Health and Social Care.

**Contributors** AA and EC drafted the manuscript. AA, EC, EC, AS, AF, JP and RB designed this study. MH was responsible for the statistical and methodological aspects of the study. AT was responsible for the economic analysis aspects of the study. The following authors lead PRESTO sites: SB (NWAS), JC (Aberdeen), KK (SWAS) and AR (WMAS). All authors contributed, read and approved the final manuscript and agree to be accountable for all aspects of the work.

**Funding** This work is supported by National Institute for Health Research grant number (PB-PG-1216-20034) and Abbott Point of Care. Also received donation of reagents from Roche Diagnostics International Ltd and LumiraDx.

**Competing interests** We received funding from Abbott Point of Care. Also received donation of reagents from Roche Diagnostics International Ltd and LumiraDx.

**Patient consent for publication** Not required.

**Provenance and peer review** Not commissioned; externally peer reviewed.

**Open access** This is an open access article distributed in accordance with the Creative Commons Attribution 4.0 Unported (CC BY 4.0) license, which permits others to copy, redistribute, remix, transform and build upon this work for any purpose, provided the original work is properly cited, a link to the licence is given, and indication of whether changes were made. See: https://creativecommons.org/licenses/by/4.0/.

**ORCID iDs**
Abdulrhman Alghamdi http://orcid.org/0000-0002-8292-5212
Eloïse Cook http://orcid.org/0000-0003-1598-4703
Aloysius Siriwardena http://orcid.org/0000-0003-2484-8201
Kim Kirby http://orcid.org/0000-0002-8092-7978

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
