## [Reviewer comments · BMJ Open]

ARTICLE DETAILS

TITLE (PROVISIONAL)	The Pre-hospital Evaluation of Sensitive Troponin (PRESTO) Study: multicentre prospective diagnostic accuracy study protocol
AUTHORS	Alghamdi, Abdulrhman; Cook, Eloïse; Carlton, Edward; Siriwardena, Aloysius; Hann, Mark; Thompson, Alexander; Foulkes, Angela; Phillips, John; Cooper, Jamie; Bell, Steve; Kirby, Kim; Rosser, Andy; Body, Richard

VERSION 1 – REVIEW

REVIEWER	Dr David C Gaze University of Westminster, United Kingdom
REVIEW RETURNED	30-Jul-2019

GENERAL COMMENTS	The authors are highly skilled in carrying out such research projects. I do not envisage any issues with the proposed PRESTO study.
---

REVIEWER	Marion A Hofmann Bowman University of Michigan, Ann Arbor, Mi, USA
REVIEW RETURNED	11-Sep-2019

GENERAL COMMENTS	4 ambulance services in the UK will participate in this study. What steps are taken to ensure that this will be a representative sample of UK practice? Specifically, are rural areas included and patients with socioeconomic diversity and racial diversity? Will the status dead/alive be checked for the patients who could not be reached for written consent within 4 weeks of initial blood draw/oral consent? It seems important to monitor for the primary outcome for patients initially included into the study, but lost for f/u of a written consent, since this group could represent a high risk group for AMI/death.
--

REVIEWER	Dennis W.T. Nilsen Stavanger University Hospital, Stavanger, and University of Bergen, Bergen, Norway.
REVIEW RETURNED	25-Sep-2019

GENERAL COMMENTS	The authors will be evaluating the diagnostic sensitivity of troponins in a prospective observational multicenter prehospital setting in the UK, recruiting patients who call emergency (999) ambulance services where the responding paramedic suspects cardiac chest pain. The primary objective of is to evaluate whether paramedics can use troponin testing by
---

	a handheld device with a computerised algorithm at the time of arrival to patients. They will be applying the "Troponin-only Manchester Acute Coronary Syndromes (T-MACS)" decision rule which is a validated tool used in the emergency department (ED). Venous blood will be drawn and stored in a lithium heparin bottle labelled with a unique study identifier and sampling will be repeated one hour (+/- 30 minutes) after the pre-hospital blood has been taken. On arrival at the hospital, all patients will undergo reference standard troponin testing in accordance with contemporary national and international guidance. The primary outcome will be a diagnosis of acute myocardial infarction (AMI), established at the time of initial hospital admission. The secondary outcomes will include any major adverse cardiac events (MACE) within 30 days of enrolment. The study will have economic implications with regard to healthcare resource use at baseline and at 30 days. The protocol is well designed!
--	--

VERSION 1 – AUTHOR RESPONSE

First point: we have added the start date and the expected end date "We started participants recruitment on February-2019, and we will complete recruitment within 12 months."

Second point: we approached all ambulances Trusts in England and Wales. This communication was facilitated by the National Ambulance Research Steering Group (NARSG). Nine of ten Ambulance Trusts in England and Wales expressed an interest in participating in PRESTO study. We are endeavouring to work with each of these Ambulance Trusts in our future programme of research. Due to limited recourses, however, we selected four Trusts that are research active, have confirmed feasibility, and covering large territories, including both urban and rural areas.

Third point: this is a very important point. Unfortunately, the conditions of ethical approval require us to destroy the samples if written consent is not obtained within 30 days. However, to mitigate the risk, we endeavour to obtain written informed consent in the Emergency Department whenever possible.

VERSION 2 – REVIEW

REVIEWER	Marion A Hofmann Bowman University of Michigan, Ann Arbor, MI USA
REVIEW RETURNED	06-Oct-2019
GENERAL COMMENTS	The reviewer completed the checklist but made no further comments.